# The identification and treatment of mental health and substance misuse problems in sexual assault services: A systematic review

Theodora Stefanidou[1], Elizabeth Hughes[2], Katherine Kester[1], Amanda Edmondson[3], Rabiya Majeed-Ariss[2], Christine Smith[3], Steven Ariss[4], Charlie Brooker[5], Gail Gilchrist[6], Sarah Kendal[2], Mike Lucock[3], Fay Maxted[7], Concetta Perot[6,8], Rebekah Shallcross[2], Kylee Trevillion[6], Brynmor Lloyd-Evans[1] *

1 Division of Psychiatry, University College London, London, United Kingdom, 2 School of Healthcare, Faculty of Medicine and Health, University of Leeds, Leeds, United Kingdom, 3 School of Human and Health Sciences, University of Huddersfield, Huddersfield, West Yorkshire, United Kingdom, 4 Centre for Assistive Technology and Connected Healthcare (CATCH) and School of Health and Related Research (ScHARR), University of Sheffield, Sheffield, United Kingdom, 5 Centre for Criminology and Sociology, Royal Holloway University of London, Surrey, United Kingdom, 6 Section of Women's Mental Health, Health Service and Population Research Department, Institute of Psychiatry, Psychology & Neuroscience, King's College London, London, United Kingdom, 7 The Survivors Trust, Rugby, Warwickshire, United Kingdom, 8 Bristol Medical School: Population Health Sciences, University of Bristol, Bristol, United Kingdom

* b.lloyd-evans@ucl.ac.uk

## Abstract

### Background

Specialist sexual assault services, which collect forensic evidence and offer holistic health-care to people following sexual assault, have been established internationally. In England, these services are called sexual assault referral centres (SARCs). Mental health and sub-stance misuse problems are common among SARC attendees, but little is known about how SARCs should address these needs. This review aims to seek and synthesise evidence regarding approaches to identification and support for mental health and substance misuse problems in SARCs and corresponding services internationally; empirical evidence regarding effective service models; and stakeholders' views and policy recommendations about optimal SARC practice.

### Methods

A systematic review was undertaken. PsycINFO, MEDLINE, IBSS and CINAHL were searched from 1975 to August 2018. A web-based search up to December 2018 was also conducted to identify government and expert guidelines on SARCs. Quality assessment and narrative synthesis were conducted.

### Results

We included 107 papers. We found that identification based on clinical judgement, support-ive counselling and referral to other services without active follow-up were the most common approaches. Evaluations of interventions for post-rape psychopathology in attendees of

**Data Availability Statement:** All relevant data are within the manuscript and its Supporting Information files.

**Funding:** This work was funded through a grant awarded to Professor Elizabeth Hughes by the UK Department of Health National Institute for Health Research (NIHR) under its Health Services and Delivery Research Programme (Reference Number: HS&DR 16/117/03). The views expressed in this paper are those of the authors and not necessarily those of the NHS, NIHR or the Department of Health. The funders and the sponsors played no role in study design, data collection, analysis, decision to publish or preparation of the manuscript.

**Competing interests:** The authors have declared that no competing interests exist.

sexual assault services provided mixed evidence of moderate quality. Very little evidence was found regarding interventions or support for substance misuse. Stakeholders emphasised the importance of accessibility, flexibility, continuity of care, in-house psychological support, staff trained in mental health as well as specialist support for LGBT groups and people with learning difficulties. Guidelines suggested that SARCs should assess for mental health and substance misuse and provide in-house emotional support, but the extent and nature of support were not clarified. Both stakeholders and guidelines recommended close partnership between sexual assault services and local counselling services.

## Conclusions

This review suggests that there is big variation in the mental health and substance misuse provision both across and within different sexual assault service models. We found no robust evidence about how sexual assault services can achieve good mental health and substance misuse outcomes for service users. Clearer guidance for service planners and commissioners, informed by robust evidence about optimal service organisations and pathways, is required.

**PROSPERO registration number**: CRD42018119706

## Introduction

### Sexual violence

Sexual violence affects many people worldwide, and most commonly women. It encompasses acts of rape, sexual assault and abuse, with the terms often used interchangeably [1]. The World Health Organisation (WHO) estimates that, globally, 35% of women have experienced physical and/or sexual violence; with most of this violence being perpetrated by their intimate partner (IPV) [2, 3]. In England and Wales, it is estimated that 25% of women and 5% of men have experienced some type of sexual violence in their lifetime [4], and the National Survey of Sexual Health Attitudes and Lifestyles (Natsal-3) [5] reports that 10% of women and 1% of men experienced non volitional sex in their lifetime. Although, women are more likely to experience sexual violence than men, those from specific populations such as men who have sex with men (MSM), military veterans, prisoners and people in hospital settings are at higher risk than men in the general population [6].

Sexual violence has a significant and often long-term impact on physical and mental health. It is associated with the development of post-traumatic stress disorder (PTSD), drug and substance misuse, self-harm, increased risk of suicide, as well as relationship problems and long-term mental illness [7–9]. Those with a history of sexual abuse have higher rates of hazardous drug and alcohol use than the general population [10], and women who experienced IPV are significantly more likely to experience adverse mental health consequences, including suicide attempts and emotional distress [1, 11]. Moreover, survivors of sexual abuse, and specifically those with a history of child sexual abuse and poor mental health, are at great risk of using sex as self-injury and being revicitmised later in life [12–14].

Trauma-focused cognitive behavioural therapies (CBT) have the strongest evidence base in addressing PTSD and other psychopathologies among sexual assault survivors [15–17]. The 2018 guidelines from the National Institute for Health and Care Excellence (NICE) recommend individual trauma-focused CBT within one month of the traumatic event for adults with

acute stress disorder or clinically important symptoms of PTSD and the use of validated screening tools [15]. People with PTSD and comorbid substance misuse should not be excluded from treatment [15], and there is meta-analytic evidence suggesting that trauma-focused therapies delivered alongside an intervention for substance misuse can be of benefit [18].

## Sexual assault services

Specialised services for people following sexual assault, which can collect forensic evidence, and address service users' physical and emotional health needs, have been established in several countries. These have various names and commissioning structures. They are commonly based in community or hospital centres, provide forensic and physical healthcare, and often link to other health and support services [19, 20].

Different models of sexual assault service provision internationally have been identified [19] and are summarised in Box 1 [19, 21, 22]. Service provision typically varies within countries, with disparity associated with an urban/rural division and differences in legislation and funding [20, 23, 24]. In England and Wales, specialist sexual assault services are called Sexual Assault Referral Centres (SARCs) and have been developed to address shortcomings in the previous medico legal response to sexual violence [20, 24]. SARCs provide holistic support; including forensic medical examinations, health checks, safeguarding and risk assessments, psychosocial support and access to services for mental health and substance misuse problems [22]. The first SARC opened at St Mary's Hospital in Manchester in 1986 [24], and since then several SARCs have been established across the country.

## Sexual assault services and mental health

Research from England suggests that approximately 40% of SARC service users are already known to mental health services and many have serious mental health problems[25, 26]. However, assessments of mental health and substance use are often not conducted in SARCs and there is a lack of clear pathways from SARCs into mental health and drug and alcohol services [25], despite policy guidance and government-provided service specifications that attendees of SARCs should have access to mental health and psychological therapies [22].

There is limited evidence about how Sexual Assault Services can best address attendees' mental health and substance misuse needs. A 2005 review [27] of the effectiveness of Sexual Assault Response Team-Sexual Assault Nurse Examiner (SANE) programs in the US found tentative evidence suggesting that SANEs are effective in promoting the psychological recovery of survivors. However, the review had significant shortcomings as all three included studies were uncontrolled with two out of three being published more than ten years ago. A comprehensive search strategy and critical appraisal of included studies were also lacking.

## Aims

To our knowledge, this is the first systematic review examining what evidence is available regarding mental health and substance misuse provision in sexual assault services, including SARCs in the UK and corresponding services internationally. To meet the current gap in knowledge, our review addressed the following three questions:

1. What are the approaches to prevention, identification and treatment of mental health and substance misuse problems in different sexual assault service models?

2. What models of treatment, service delivery and organisation in sexual assault services are effective regarding service users' mental health and substance misuse outcomes?

> ## Box 1: International models of specialist sexual assault services
>
> Sexual assault referral centres (SARCs) are the main providers of sexual assault services in the UK. SARCs are based in a hospital or community setting and provide holistic support, including forensic medical examinations, health checks, safeguarding and risk assessments, psychosocial support and access to services for mental health and substance misuse problems.
>
> The sexual assault response team-sexual assault nurse examiner (SART-SANE) model is the predominant model in the US and has been modelled in Australia, Canada and the UK. The SANE programs consist of specially trained forensic nurses that provide clinical and forensic support to the victim. SANEs usually operate as part of interdisciplinary teams known as SARTs.
>
> Sexual assault/domestic violence care and treatment centres (SA/DVCTCs) are hospital based centres that provide 24/7 support to victims of sexual and domestic violence in Canada.
>
> In Australia, specialist services for victims are provided by the sexual assault resource teams (SARC) in the Western part of the country, and the sexual assault services (SAS) in New South Wales.
>
> The specialist sexual violence services (SSVS) are non-government organisations that consist the main provider of specialist services to victims of sexual violence in New Zealand.
>
> Sexual assault centres (SACs) aim to provide comprehensive care to victims and can be found in most Nordic countries.
>
> Thuthuzela care centres (TCCs) are 'One-Stop Shop' centres in hospitals or communities that provide healthcare and forensic support to victims of rape in South Africa.
>
> The Child Advocacy Centre (CAC) model has been adopted in the US and Europe and provides holistic support at a stand-alone facility to children that have been sexually assaulted.

3. What are stakeholders' views and policy recommendations about how sexual assault services should prevent, identify and treat mental health and substance misuse problems for people following a sexual assault?

The review follows guidance from the Centre for Reviews and Dissemination on undertaking reviews in health care [28] and the Preferred Reporting Items for Systematic Reviews and Meta-analyses (PRISMA statement) [29]. A PRISMA checklist for this review is provided in the data supplement (S1 File).

## Materials and methods

### Protocol and registration

The review protocol was registered at PROSPERO international prospective register of systematic reviews at the Centre for Reviews and Dissemination, University of York (CRD42018119706) [30].

## Inclusion criteria

We included studies, which met the following criteria:

a.  Services: We included studies of sexual assault services, if they offered (1) specialist support, within a single service, to adults and/or children who have experienced sexual assault, and (2) provided healthcare provision and the collection of forensic evidence and legal statements. We excluded services that did not fulfil both these functions, even if they provided some support to people who had experienced sexual assault.

b.  Publication types: Both peer-reviewed and grey literature were eligible.

    1.  For our first review question, we included publications that provided any information about how a sexual assault service prevented, identified or treated mental health and substance misuse problems, whether or not this was the main focus of the paper. Journal articles, PhD theses, conference proceedings and book chapters were eligible for inclusion.

    2.  For our second review question, we included quantitative comparison studies of any design, which compared sexual assault services with standard care; different sexual assault service models; or two different interventions/packages of care within a sexual assault service.

    3.  For our third review question, we included qualitative interviews, focus groups or surveys of sexual assault service stakeholders. These included any report of participants' views or recommendations regarding the prevention, identification and treatment of mental health and substance misuse problems in sexual assault services. We also considered guidelines from government or expert bodies specifying mental health and substance misuse provision in the UK SARC model only.

c.  Participants: Not applicable to review question one. Review question two: adult and/or child service users of sexual assault services. Review question three: Sexual assault services 'stakeholders including service users, staff and other professionals working closely with sexual assault services.

d.  Intervention: Not applicable to review question one and three. For review question two, the intervention could be a sexual assault service; a sexual assault service model; an intervention/package of care within a sexual assault service designed to improve mental health and substance misuse outcomes.

e.  Comparison: Not applicable to review question one and three. For review question two, the comparison could be standard care, however defined; a different sexual assault service model; or a different intervention/ package of care within a sexual assault service designed to improve mental health and substance misuse outcomes.

f.  Outcomes: Not applicable to review question one and three. Review question two: any measure evaluating mental health, and substance misuse outcomes at end of sexual assault service treatment or longer-term follow-up; and/or use mental health inpatient or crisis services at follow-up.

Due to limited resources, we excluded literature written in languages other than English if an English version was not available.

## Search strategy

Electronic searches were conducted on the following bibliographic databases: PsycINFO, MEDLINE, IBSS and CINAHL, combining text words for SARCs and sexual assault services adapted from an international overview of sexual violence services [19] (S2 File). We consulted a healthcare librarian and piloted our search to ensure all key studies were retrieved. Relevant medical subheadings (MeSH terms) were not identified, and so not used in our search. To ensure all relevant records were retrieved, we did not limit the searches by adding mental health and substance misuse terms. We restricted our search to humans and we set 1975 as the lower date restriction because the SANE model was introduced in the mid-1970s, and the SARC model was introduced in 1986 [20, 27]. The last search was conducted in August 2018.

We searched grey literature and conference proceedings in Google Scholar and Zetoc databases, and for SARC policy guidance in the English Primary Care Commissioning, Department of Health and Social Care and NHS England websites. The last web-based search was conducted in December 2018. Finally, we consulted experts in the field to identify any additional records.

Two members of the review team (TS, AE) piloted the selection strategy. All retrieved records were independently screened by two reviewers (TS, CS and AE, KK). Any disagreements were resolved by consensus, and when necessary by involving a third reviewer (BLE). Title and abstract screening was carried out in Endnote X8, and full-text screening in Covidence.

## Data extraction

A data extraction schedule was adapted from the Cochrane Collaboration data collection form for intervention reviews: RCTs and non-RCTs, and was piloted by two member of the team (TS, AE). Data extraction was undertaken by a member of the team (CB, KT, GG, SA, EH, AE, CS, SK, RMA, BLE, DS) with 10% of the extraction being checked by another team member (TS). For review question one, we retrieved information about first author, year of publication, country, study design, setting, and mental health and substance misuse provision in sexual assault services. For review question two, data were extracted on first author, year of publication, country, study design, number of participants, setting, service characteristics, evaluated outcomes, and results. For review question three, we extracted the study characteristics as above, and the main findings/recommendations about mental health and substance misuse provision in sexual assault services. We contacted authors for any information not available in the published papers.

## Quality assessment

The quality of included studies for review questions two and three was assessed by a member of the team (TS, KK or RMA) using the Mixed Methods Appraisal Tool (MMAT) [31]. This is an established assessment tool, which can be used with quantitative, qualitative and mixed methods primary studies. The MMAT quality score ranges from zero (low quality) to five (high quality). Quality scores were reported and considered in the narrative synthesis of the evidence, but papers were not excluded on account of low-quality scores.

## Data synthesis

A narrative synthesis was undertaken, following ESRC guidelines [32]. For review question one, we first identified papers that described actual arrangements within specific sexual assault services and extracted data from these. Information pertaining to mental health and substance misuse was initially extracted verbatim, then summarised. Following team discussion, we

distinguished three domains of mental health and substance misuse approaches (identification, support and referral) and, with reference to an existing classification of sexual assault services [19], we identified from included papers different types of service provision within each domain. Lastly, we explored commonalities and variations in practice regarding mental health and substance misuse between and within different sexual assault service models.

For review question two, we described the sexual assault services' interventions, participants, study design and outcomes. Due to anticipated heterogeneity in all these fields, quantitative synthesis of results was not considered. In a narrative summary, we considered the consistency and strength of evidence regarding the effectiveness of different sexual assault service models and interventions.

For review question three, we summarised views or recommendations made by sexual assault stakeholders and SARC policy makers: i) directly about identification or support with mental health or substance misuse; and ii) about more general features of service provision with relevance to mental health and substance misuse. We considered differences in perspective between stakeholder groups and policy sources.

Finally, we compared and synthesised findings from all three review questions. We considered the overall evidence about how to address mental health and substance misuse needs in sexual assault services with reference to the extent and quality of the empirical evidence, and the degree of consistency in findings from interviews with stakeholders and policy guidelines.

## Results

### Study selection

The bibliographic database search yielded 3,774 citations and the web-based search yielded 965. Another 12 records were identified through consultation with experts. After duplicates were removed, abstracts and summaries from 3,957 publications were screened for eligibility. After screening, 107 full-text papers were included in the review. A PRISMA diagram describing study screening and selection is provided in Fig 1.

### Study characteristics

Of the 107 papers included in the review, some were relevant to more than one of our three review questions. For review question one, we included 78 publications providing information on how sexual assault services address mental health and substance misuse [20, 24–26, 33–106]; for review question two, we included 6 papers of five trial evaluations [84, 89, 92, 107–109]; and for review question three, we included 34 studies on stakeholders' views [20, 25, 34, 37, 44, 53, 54, 60, 62, 72, 78, 82, 86, 90, 93, 94, 97, 110–126] and 10 policy documents [20, 22, 127–134]. Papers were published between 1979 and 2019 and covered services around the world.

Quality assessment was conducted for evaluation studies included in review question two, and qualitative studies included in review question three. The median MMAT quality score for evaluation studies was three, which indicates studies of moderate quality. Five of these obtained a score of three (moderate quality), and one study obtained a score of four (moderately high quality). The main quality concerns were related to unreported or poorly performed randomisation, and low follow-up rates. Studies of stakeholders' views had a median MMAT quality score of five, which indicates studies of comparatively high methodological quality. Twenty-one of these obtained a score of five (high quality), eight a score of four (moderately high quality), two a score of three (moderate quality), and three studies scored two or less (low quality). A breakdown of the MMAT scores for each study is reported in the data supplement (S3 File)

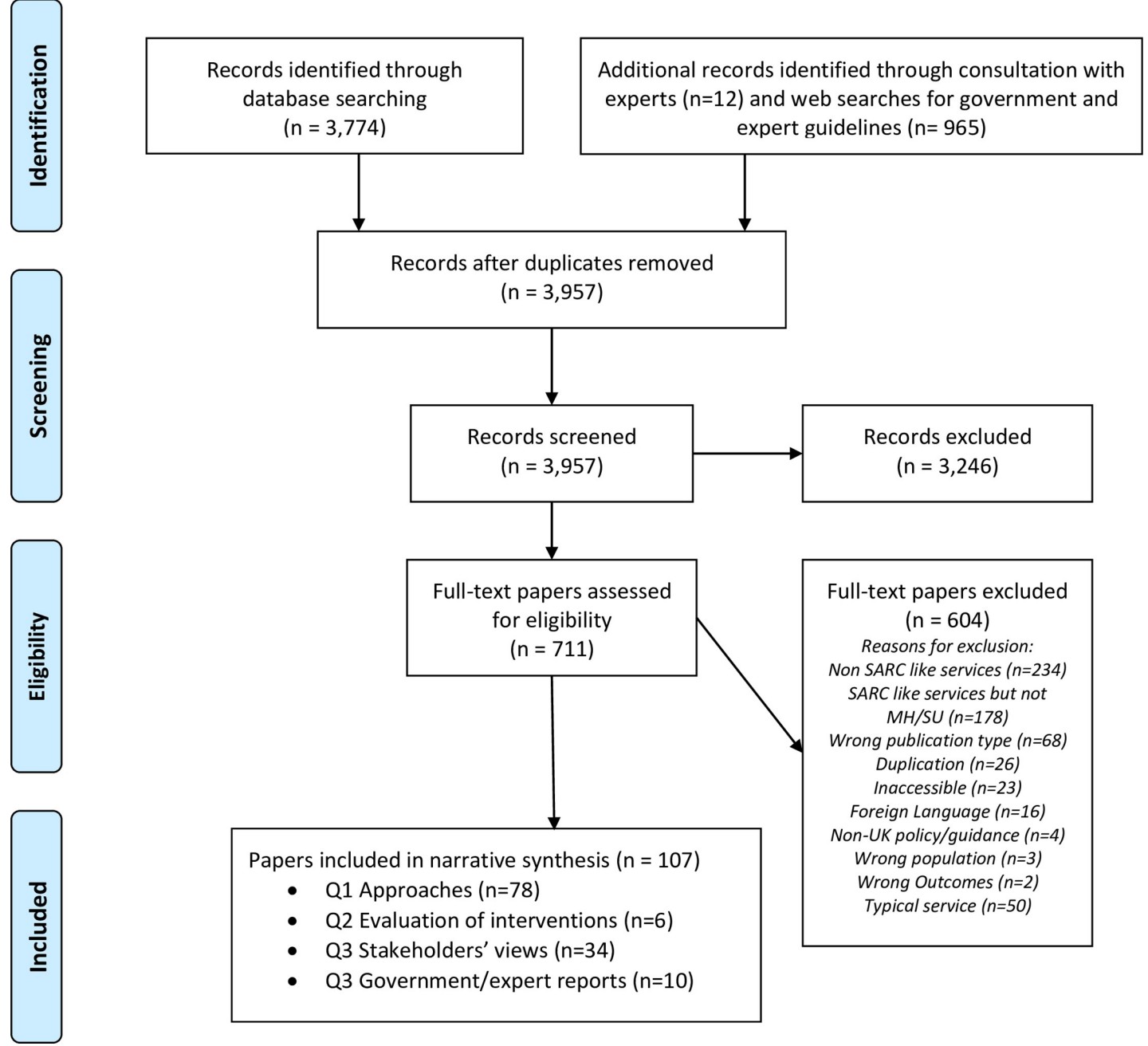

**Fig 1. PRISMA 2009 flow diagram.**

## Q1: Approaches to mental health and substance misuse

We found 78 papers [20, 24–26, 33–106] published between 1979 and 2019 that provided information on how sexual assault services identify and treat mental health and substance misuse problems. Of these, 64 were journal articles [24–26, 33, 34, 36, 38, 39, 41–58, 60, 61, 63–67, 69–85, 88–92, 94–96, 98–100, 102, 105, 106], and 14 were retrieved through grey literature searches [20, 35, 37, 40, 59, 62, 68, 86, 87, 93, 97, 101, 103, 104]. The included papers described services from the UK, US, Canada, Australia, New Zealand, Nordic Countries, Africa, Turkey,

**Table 1. Approaches to mental health and substance misuse by service model.**

| Included papers (N = 78) | Identification (N = 32) | | | In-house support (N = 57) | | | | | Referral-on (N = 39) | |
|---|---|---|---|---|---|---|---|---|---|---|
| | Unstructured/ unspecified assessment (N = 14) | Structured assessment (not using validated measures) (N = 10) | Structured assessment (using validated standardised measures) (N = 5) | Immediate emotional support only at first contact (N = 2) | Follow-up care: counselling (N = 32) | Follow-up care: structured psychological interventions (N = 5) | Unspecified type of emotional support (N = 7) | No provision of emotional support (N = 2) | Signposting and referral (n = 32) | Referral on with active follow-up (n = 6) |
| SARC (UK) N = 22 | 5 | 3 | 0 | 0 | 7 | 0 | 4 | 1 | 9 | 2 |
| SANE-SART (US) N = 11 | 5 | 0 | 0 | 0 | 6 | 0 | 0 | 0 | 5 | 3 |
| SA/DVCTC (Canada) N = 3 | 1 | 1 | 0 | 1 | 2 | 0 | 0 | 0 | 2 | 0 |
| SARC/SAS (Australia) N = 6 | 0 | 1 | 0 | 0 | 4 | 1 | 0 | 0 | 0 | 0 |
| SAC (New Zealand) N = 1 | 0 | 0 | 0 | 0 | 0 | 1 | 0 | 0 | 0 | 0 |
| SAC (Nordic countries) N = 8 | 1 | 0 | 1 | 0 | 3 | 1 | 0 | 0 | 1 | 0 |
| TCC (South Africa) N = 1 | 0 | 0 | 0 | 0 | 0 | 0 | 1 | 0 | 1 | 0 |
| CAC N = 12 | 1 | 3 | 1 | 1 | 4 | 2 | 0 | 0 | 7 | 1 |
| Other services N = 14 | 1 | 2 | 3 | 0 | 6 | 0 | 2 | 1 | 7 | 0 |

SARC = sexual assault referral centre; SANE-SART = sexual assault nurse examiner-sexual assault response team; SA/DVCTC = sexual assault/domestic violence care and treatment centre; SARC/SAS = sexual assault resource team/ sexual assault service; SAC = sexual assault centre; TCC = Thuthuzela care centre; CAC = child advocacy centre

*Note: We did not double count papers describing similar service provision within the same service. Specifically, we found 2 separate papers describing similar service provision in the SAC in St Olav's Hospital in Oslo, Norway [65, 102], 4 separate papers describing similar provision in the Havens in London, UK [42, 67] [74, 95], 6 separate papers describing similar service provision in the St Mary's SARC in Manchester, UK [20, 55, 59, 81, 83, 90], 2 separate papers describing similar service provision in a SARC in Perth, Australia [47, 50], 3 separate papers describing similar service provision in the Copenhagen sexual assault centre in Denmark [76, 96, 98]. 'Other services' included Child and Youth Protection Programs (CYPPs) in Canada, individual centres for sexual assault in Zambia, Israel, Hawaii, Kenya, US, Sierra Leone, Congo; a centre for sexual and family violence in the Netherlands; and sexual assault services in South Africa that did not operate as TCCs at the time of publication.

Israel and Hawaii. The identified mental health and substance misuse approaches are summarised by service model in Table 1. Services that we were unable to assign to a specific service model are grouped as "other services". See S4 File for a full list of included studies and description of the reported approaches to mental health and substance misuse within each sexual assault service model.

**Identification.** 32 papers provided information on how sexual assault services identify mental health and substance misuse problems among service users [25, 26, 35–40, 42, 45, 48, 49, 52, 54, 56–58, 64, 65, 67, 80, 81, 84, 86, 87, 93, 95, 101, 104–106]. Some sexual assault services were described in two or more papers: our 32 included papers described 29 different sexual assault services. Different approaches were:

1. **Unstructured or unspecified assessment:** Sexual assault services were reported as seeking to identify mental health and substance misuse problems, but the methods used were either unspecified or unstructured. Unstructured approaches included using professional judgement, casual observation, medical history taking, relying on self-reported disclosure, and one-off questions about mental health. Unstructured/unspecified assessment was the most commonly reported identification method (n = 14) across all included services [25, 26, 35, 36, 38–40, 48, 49, 56, 84, 93, 101, 104], especially in SARCs in the UK and SANE-SARTs in the US.

2. **Structured assessment (not using validated measures),** where information on mental health and substance misuse was gathered consistently using a standard set of questions, but this was not reported to involve the use of a validated assessment or screening tool(s). The structured assessment approach mainly involved assessing: self-harm, suicidal ideation, PTSD, and mental health history. This was the second most commonly reported identification approach (n = 10) and it was found to be the most common method in CACs in Turkey and the US, and Australian SARCs/SASs [37, 54, 58, 64, 67, 80, 81, 86, 87, 106].

3. **Structured assessment using validated standardised measures,** where information on mental health and substance misuse was gathered consistently through a structured assessment using validated screening tools. Identified measures used in services were: UCLA PTSD trauma screen [135], Trauma Symptom Checklist for Children (TSCC) [136], Trauma Screening Questionnaire (TSQ) [137], Children's Revised Impact of Event Scale (CRIES) [138], strengths and difficulties questionnaire [139], a validated measure of problem substance use based on the DSM-IV criteria, Child's Report of Parenting Behaviour Inventory [140]. The use of validated measures was the least commonly described approach (n = 5), only reported in CACs, Nordic SACs, and Other services [45, 52, 57, 65, 105].

There was also variation on the time point at which services screened for mental health and substance misuse. Some were reported to assess risk and obtain mental health history at the first contact with the service, others at a later follow-up point, and others at both times. We found more variation within than between service models.

**In-house support.**   57 papers describing 48 different sexual assault services described in-house support for mental health and substance misuse problems [20, 24, 26, 33–42, 46, 47, 50, 51, 53–56, 58, 59, 61–63, 65–68, 70, 72–78, 81, 83, 85, 87, 89–99, 101, 103–105]. Reported approaches were categorised as: i) immediate emotional support only at first contact, ii) follow-up care: counselling, iii) follow-up care: structured psychological interventions, iv) unspecified type of emotional support, v) no emotional support provided.

The most reported type of support across all service models (except TCC) was the provision of counselling as part of the follow-up care (n = 32) [20, 24, 26, 34–36, 38, 40–42, 46, 47, 50, 51, 53, 55, 56, 58, 59, 65–68, 74–78, 81, 83, 85, 87, 89–91, 94–98, 101, 103]. We grouped as counselling any type of counselling that was not specified as a structured psychological therapy. That included crisis, supportive and unspecified types of counselling provided by a member of the sexual assault team. The provision of structured psychological interventions as part of the in-house follow-up care was reported only in five papers specifying services in Nordic countries, Australia, New Zealand and CACs [37, 39, 63, 73, 99]. These interventions included an adjunct equine facilitated therapy program, cognitive behaviour therapy (CBT), eye movement desensitisation reprocessing (EMDR) therapy and art therapy. Two papers [54, 92] detailing a Canadian Service and a CAC in the US described the provision of immediate emotional support only. Two papers [33, 62] described services that did not provide any type of in-house emotional support, and seven papers [33, 61, 70, 72, 93, 104, 105]reported the provision of

emotional support within the sexual assault service, but with insufficient information provided within the paper or subsequently by authors for us to categorise it as any particular type.

**Referral-on.** 39 papers describing 38 different sexual assault services described arrangements for referring service users on from the sexual assault services to other mental health and substance misuse services [20, 33, 35–38, 40, 42–44, 49, 51, 53, 54, 56, 58, 60–62, 66, 67, 69, 71, 72, 74, 79, 82, 85, 86, 88, 92, 93, 95, 97, 100, 101, 103–105]. Most reported that the sexual assault service staff provided service users with information on available follow-up services and made appropriate referrals to services outside of the sexual assault service. Only six papers [20, 49, 60, 71, 95, 101] reported services within the UK SARCs, SANE-SARTs and CACs models that referred on with active follow-up. These services followed up people after the referral to another service had been made to determine if the person has accessed these services and to check ongoing or emerging mental health and substance misuse needs.

Overall, we found wide variation in approaches both between and within different sexual assault service models for all three domains. Unstructured assessment, unspecified counselling and referral on without assertive follow up are the most common approaches, with a minority of services providing more structured/specialist provision within each domain.

## Q2: Evaluation of interventions

Six papers [84, 89, 92, 107–109] reported findings from five studies, evaluating the effectiveness of interventions for mental health or substance misuse within a sexual assault service. All six papers were peer-reviewed and published between 2003 and 2017. All studies were randomised controlled trials (RCTs) with four conducted in the US and one in Australia. Outcomes assessed were substance use and abuse, anxiety, depression and PTSD. Characteristics and key findings of each study are summarised in Table 2.

Five papers [84, 92, 107–109]reported four trials of brief video interventions (VIs) to prevent post-rape psychopathology, implemented before or following a medico-legal examination. The VIs were delivered as an addition to standard sexual assault service care, which included the completion of a medico-legal examination and in most cases an additional meeting with a rape crisis counsellor/advocate. The videos were between nine to twenty minutes in length. In three studies [92, 107, 109], the video was shown before the medico-legal examination, and included information on what would happen during the examination and psycho-educational material about self-management strategies to prevent post-assault distress and/or substance use. In one study [84], the video was shown after the medico-legal examination and involved just the psycho-educational material about self-management strategies. The trial follow-up periods varied from 2 weeks to 6 months.

Two papers [107, 108]reported different findings from the same study, which evaluated a seventeen minute two-part VI offered immediately before the medico-legal examination (n = 225; n = 226). One reported a positive effect for the intervention on risk of marijuana abuse (one of six drug and alcohol use outcomes evaluated) at six weeks follow-up [107]. The other [108] found positive effects of the intervention on PTSD and depression among those with a prior history of assault, but there were no significant differences between groups on any outcomes across the whole sample. The VI evaluated by Miller 2005 [84] evaluated a nine minute VI shown post examination and found a significant positive effect for the intervention group compared to usual care on anxiety at two months follow-up(n = 179). Across the whole sample, there was no difference between groups in PTSD symptoms or self-reported distress. The other two studies testing VIs [92, 109] found no effects on any MH/SM outcomes evaluated (n = 69; n = 245). One of them was a pilot study [92], and as such likely underpowered to detect significant differences between groups. Follow-up rates at the final evaluation time

**Table 2. Evaluation of interventions–Study characteristics and outcomes.**

| Study Reference | Setting | Study design | Participants | Intervention Group | Control Group | Follow up rate (%, n/N) | MH/SM outcome measures | Main findings | MMAT score |
|---|---|---|---|---|---|---|---|---|---|
| Acierno (2003) [107] | Southeastern academic medical centre, US | RCT | Female(ages 15+) victims of recent sexual assault (N = 226) | Video Intervention: standard care plus a 17-min video shown immediately pre-examination (n = 117) | Standard care: Medico-legal examination and presence of a rape crisis counsellor during the examination (n = 109) | 6 weeks follow-up 55% (124/226) | Used self-report questionnaires to measure alcohol use and abuse, marijuana use and abuse, "hard drug" use and abuse | Found reduced risk of marijuana abuse for the VI group at 6 weeks follow-up (p = 0.046). No significant differences between groups on other outcomes. | 3 |
| Resnick (2007) [108] | Southeastern academic medical centre, US | RCT | Female(ages 15+) victims of recent sexual assault (N = 225) | Video Intervention: standard care plus a 17-min video shown immediately pre-examination (n = 117) | Standard care: Medico-legal examination and presence of a rape crisis counsellor during the examination (n = 108) | 6 weeks follow-up 55% (123/225) 6 months follow-up 57% (128/225) | i)Post-traumatic Symptoms Scale (PSS-SR) ii)Beck Depression Inventory (BDI) iii)Beck Anxiety Inventory (BAI) | Descriptive data for treatment and control groups at baseline and follow-ups, and statistical comparisons between the two groups were not clearly reported. Across the whole, sample no significant differences between groups on any outcomes. | 3 |
| Miller (2015) [84] | SANE program at a hospital, South Carolina, US | RCT | Female (ages 18+) victims of recent sexual assault (N = 179) | Video Intervention: Standard care plus a 9-min psychoeducation video shown post-examination (n = 94) | Standard care: SANE medico-legal examination and meeting with rape crisis advocate (n = 85) | 2 weeks follow-up 39%(69/179) 2 months follow-up 41% (74/179) | i) Subjective Units of Distress (SUDs) ii) PTSD Symptom Scale (PSS-SR) iii) State Trait Anxiety Inventory (STAI) | Found fewer anxiety symptoms in VI group at 2 weeks and 2 months follow-up (p<0.05).No significant differences between groups on other outcomes. | 3 |
| Nixon (2017) [89] | Yarrow Place Rape and Sexual Assault Centre, Adelaide, Australia | RCT | Adult (ages 18 +) victims of sexual assault within the last month; meeting criteria for Acute Stress Disorder (ASD) (N = 47) | Cognitive Processing Therapy (CPT): 6 weekly, 90 minute sessions focused on cognitive restructuring and trauma processing (n = 25) | TAU: non-CBT supportive counselling (i.e. interpersonal therapy, mindfulness, etc.) (n = 22) | 3 months follow-up 51% (24/47) 6 months follow-up 51% (24/47) 12 months follow-up 53% (25/47) | Primary outcomes Acute stress disorder (ASD) and PTSD diagnosis and symptom severity (CAPS; PCL-S; PTCI) Secondary outcomes Comorbid mood, other anxiety and substance use disorders (MINI; BDI-II) | Found no significant differences between groups on any outcomes. | 3 |

*(Continued)*

**Table 2.** (Continued)

| Study Reference | Setting | Study design | Participants | Intervention Group | Control Group | Follow up rate (%, n/N) | MH/SM outcome measures | Main findings | MMAT score |
|---|---|---|---|---|---|---|---|---|---|
| Rheingold (2013) [92] | Child Advocacy Centre, South East region of the US | Pilot RCT | Children (ages 4–15) victims of sexual assault and respective care-givers (N = 69) | Video Intervention: Standard care plus a 20-min psychoeducation video shown pre-examination both to care-giver and child (n = 35) | Standard care: Child sexual assault medical examination (CSAME) and support from rape crisis victim advocate (n = 34) | 6 weeks follow-up Children = 26% (18/69) Caregivers = 45% (31/69) | Child i)STAI anxiety inventory ii) Subjective units of distress (SUD) iii)Beck Anxiety Inventory (BAI) iv)Trauma Symptoms Checklist for Children (TSCC) Care-giver i) STAI anxiety inventory ii) Subjective Units of Distress (SUD) iii) Caregiver report of distress during examination | Found the intervention to be feasible and acceptable. No significant differences between groups on any outcomes. | 3 |
| Walsh (2017) [109] | 2 medical centres with SANE program, Midwest Metropolitan area, US | RCT | Female (ages 15+) victims of recent sexual assault (N = 245) | Brief Prevention of Post-Rape Stress (PPRS) Video Intervention: standard care plus a 9-min video delivered in a single session immediately pre-examination (n = 82) | Standard care: Sexual assault medico-legal examination (n = 81) Active Control: 9-minute video providing guided muscle relation and breathing exercises (n = 82) | 2 months follow-up 63%(154/245) 3.5 months follow-up 55%(135/245) 6.5 months follow-up 49%(121/245) | i) Self-reported alcohol use ii) Alcohol Use Disorders Identification Test (AUDIT) iii) Self-reported marijuana use ii) Drug Abuse Screening Test (DAST) | Found no significant differences between groups on drug or alcohol use or abuse. | 4 |

RCT = Randomised controlled trial; SANE = Sexual assault nurse examiner

*Acierno (2003) and Resnick (2007) reported findings on different outcomes from the same study

point were low, varying across studies from 39% to 57%. It was not reported whether the studies were adequately powered to detect differences between groups.

One RCT [89] evaluated a six-session "Cognitive Processing Therapy" cognitive behavioural intervention at a sexual assault service in Adelaide, Australia. It was provided as an addition to routine care which included non-CBT supportive counselling, in addition to health care and collection of forensic evidence. None of the mental health outcomes was significant in this small trial (n = 47), which is likely to have been underpowered to test effectiveness.

## Q3: Studies on stakeholders' views

We found 34 papers [20, 25, 34, 37, 44, 53, 54, 60, 62, 72, 78, 82, 86, 90, 93, 94, 97, 110–126] relating to 32 studies, which reported stakeholders' views on how sexual assault services should identify and respond to mental health and substance misuse problems for people following sexual assault. Twenty five were peer-reviewed publications [25, 34, 44, 53, 54, 60, 72, 78, 82, 90, 94, 110–123] and nine identified through grey literature searches [20, 37, 62, 86, 93, 97, 124–126]. These were published between 1980 and 2018. Thirteen studies were set in the US, twelve in the UK, four in Canada, two in South Africa and one in seven European countries. Eleven studies included staff from sexual assault services as participants, five included service

 

users, two included carers, one included staff from partner agencies working closely with sexual assault services, and thirteen included a mix of stakeholders. The included studies consisted of surveys, interviews and quantitative questionnaires, and are summarised in Table 3.

Eighteen studies [20, 25, 37, 54, 60, 62, 78, 86, 90, 94, 97, 111, 113, 117, 118, 121, 122, 124] reported clear preferences or recommendations by stakeholders specifically regarding assessment and management of mental health and substance misuse problems. Stakeholders suggested that sexual assault services should systematically assess for mental health and substance misuse problems, and provide in-house counselling for both adult and child service users [20, 37, 54, 62, 86, 94, 97, 111, 117]. Counselling should be provided for as long as needed and should be flexible to accommodate service users' needs (i.e. gender of counsellor, appointment times etc.) [20, 37]. Staff in sexual assault services should receive training on how to assess and manage mental health and substance misuse problems, including training on how to support subgroups like people with learning difficulties [86, 90, 111]. Stakeholders emphasised the need for clear and prompt referral pathways as well as closer links with local mental health and substance misuse services [25, 86, 113, 118, 124]. They recommended that staff in sexual assault services should not just refer to other services, but also facilitate and encourage the take-up of follow-up services [20, 97, 121, 122]. Most commonly reported recommendations are summarised in Table 4.

Twenty five studies [20, 34, 37, 53, 62, 72, 86, 93, 94, 97, 110, 112–116, 118–126] reported more general perspectives and recommendations by stakeholders regarding service organisation or delivery within sexual assault services, with relevance to the experience of people with mental health and substance misuse needs. Characteristics and recommendations of each study are provided in the data supplement (S5 File).

Stakeholders highlighted the importance of making sexual assault services more widely accessible. They suggested a 24/7 access, multi-lingual service that is flexible about timing and location of appointments [37, 53, 93, 124–126]. Sexual assault services should also actively publicise their service and advertise it as clearly available to specific groups like LGBT people to overcome some of the access barriers [34, 53, 113, 125]. Regarding treatment, sexual assault services should offer a "one-stop-shop" multi-agency service that provides holistic support. Service users' social and emotional needs should be addressed as well as medical and forensic needs [20, 86, 93, 94, 113, 116, 118, 121–124]. A need was identified for specialist support within sexual assault services for LGBT clients [86, 125].

Stakeholders suggested that sexual assault services should work with referrers to minimise service users having to repeat their story on multiple occasions [93, 94]. Sexual assault services must ensure service users know they can decline a forensic examination/assessment and still receive support [34, 94, 120]. Other recommendations included that: i) staff in sexual assault services should work closely with age-related organisations to facilitate support for older survivors of sexual assault [110]; ii) support to female service users should be provided by female staff [20, 34, 97, 121, 124]; iii) s staff in sexual assault services should always display kindness and compassion and offer unconditional acceptance and belief of the service users [20, 93, 114, 115, 121, 122]; iv) Service users should be clearly informed about what are the sexual assault service procedures and should have choice and control over what happens to them at all times [20, 34, 44, 97, 113, 114].

## Q3: Government and expert guidelines

We found ten documents of government and expert guidelines [20, 22, 127–134] making recommendations about the mental health and substance misuse provision in SARCs in England. The documents included six UK government documents [22, 127, 131–134], three European

**Table 3. Stakeholders' views—Study characteristics.**

| Study reference | Setting | Stakeholder group | N | Study type | MMAT score |
|---|---|---|---|---|---|
| Ahrens (2000) [34] | 2 SANE programs, Michigan, US | SANE staff | n/s | Qualitative interviews + review of service documents | 5 |
| Belew (2012) [37] | Chicago Children's Advocacy Centre (CCAC), US | Staff (mental health therapists working within 5 Child Advocacy Centres) | n = 5 | Online questionnaire survey, including free text questions | 3 |
| Bows (2018) [110] | Various sites, England, UK | Mixed stakeholders (SARC staff and other professional stakeholders from various services including rape crisis, domestic violence organisations, and services that work with male survivors) | n = 23 | Qualitative interviews (5 out of 23 were SARC staff) | 5 |
| Brooker (2015) [25] | 25 unspecified SARCs, England, UK | SARC staff | n = 25 | Online questionnaire survey | 2 |
| Brooker (2018) [111] | England, UK (national) | SARC staff (Forensic physicians) | n = 45 | Online questionnaire survey | 4 |
| Burton (2002) [112] | CAC, Kentucky, US | Other professionals (local GPs working with Child Advocacy Centres) | n = 7 | Written questionnaire survey | 4 |
| Campbell (1998) [113] | 22 unspecified sites, US | Staff (21/22 sites including a SART program) and adult SART service users (5 sites) | n/s | Qualitative interviews (with staff from all 22 programs and service users from 5 programs) | 5 |
| Campbell (2005; 2006) [44, 115] | Unspecified SANE programs, US | SANE staff | n = 110 | Structured (telephone) interviews | 4; 5 |
| Campbell (2013) [114] | 2 Midwestern SANE programs, US | SANE service users, female, age 14–17 (from two SANE programs) | n = 20 | Qualitative interviews | 5 |
| Clark (1998) [116] | Child Sexual Abuse Response Team (CSART), Georgia, US | CSART staff | n = 13 | Qualitative interviews | 3 |
| Cole (2007; 2008) [117, 126] | SARTs, Kentucky, US | SART staff (SART program staff n = 31) and other professional stakeholders (n = 48) | n = 79 | Telephone questionnaire survey | 5 |
| COSAI (2012) [124] | 7 European countries | Mixed stakeholders (sexual assault service' staff and other professional stakeholders from 7 countries, four with SARC-type services) | n = 22 | Qualitative telephone interviews | 2 |
| Cowley (2014) [118] | SANEs, England, UK | SANE nurses | n = 5 | Qualitative interviews | 5 |
| Downing (2012) [119] | SANEs, Iowa, US | SANE staff ( | n = 14 | Qualitative interviews | 5 |
| Du Mont (2004) [54] | 15 Sexual Assault Care and Treatment Centres(SACTCs), Ontario, Canada | SACTCS staff (physicians working in Sexual Assault Care and Treatment Centres) | n = 31 | Written questionnaire survey | 4 |
| Du Mont (2009) [120] | 30 SADVTCs, Ontario, Canada | Service users (adult women using Sexual Assault/ Domestic Violence Treatment Centres) | n = 19 | Qualitative interviews | 5 |
| Du Mont (2014) [53] | 30 unspecified SA/DVTCs, Ontario, Canada | Service users (using Sexual Assault / Domestic Violence Treatment Centres) | n = 993 | Written questionnaire survey | 4 |
| Ericksen (2002) [121] | Specialised sexual assault service, British Columbia, Canada | Service users (adult women using a SANE program in a hospital emergency department) | n = 8 | Qualitative interviews | 5 |
| Fong (2016)[1] [60] | Philadelphia, Children's Alliance Child Advocacy, US | Family carers of service users (caregivers of children under 13 using "Children's Alliance") | n = 22 | Qualitative interviews | 5 |
| Goddard (2015)[2] [62] | The Havens, London, England | Mixed stakeholders (SARC staff from the Havens and other professionals including doctors, CCCG children's commissioners, CAMHS teams, third sector providers, nurses) | unclear | Online surveys and structured interview questionnaires | 4 |
| Harvey (2014)[3] [125] | Wales, UK (Various sites) | Mixed stakeholders (SARC staff and other professional stakeholders from various services across the domestic and sexual violence, and LGBT sectors) | n = 18 | Qualitative telephone interviews | 5 |
| Holton 2018 [72] | Eden District, South Africa (3 government hospitals) | Service users (adults, children, adolescents) | n = 10 | Qualitative interviews | 5 |

(*Continued*)

**Table 3.** (Continued)

| Study reference | Setting | Stakeholder group | N | Study type | MMAT score |
|---|---|---|---|---|---|
| Lippert (2008)[4] [78] | The Dallas Children's Advocacy Centre, US | Family carers of service users (caregivers of children using a Child Advocacy Centre) | n = 45 | Qualitative interviews | 4 |
| Lovett (2004)[5] [20] | 4 SARCs, Northern England, UK | SARC service users | n = 49 | Qualitative interviews | 5 |
| | | SARC staff and other professional stakeholders (police, prosecutors and victim support staff) | n = 110 | | |
| Maier (2012) [122] | SANEs from 4 "East coast states", US | SANE staff | n = 40 | Qualitative interviews | 5 |
| Mathews (2013) [82] | Two dedicated sexual assault centres in Cape Town, Western Province, South Africa | Service users (n = 30 girls age 8–17) and family care-givers (n = 30) | n = 60 | Qualitative interviews | 5 |
| Musgrave (2014) [86] | The Ferns centre in Suffolk and the Harbour Centre in Norfolk, SARCs, England, UK | SARC staff and other professional stakeholders (GUM clinic doctors, police, voluntary sector counsellors) | n = 31 | Qualitative face-to-face and phone interviews | 5 |
| Olsen (2017) [90] | St Mary's SARC, Manchester, England, UK | SARC staff | n = 42 | Online questionnaire survey | 5 |
| Robinson (2009)[6] [93] | Ynys Saff SARC, Cardiff, Wales, UK | Other professionals (senior police, health and voluntary agencies staff planning a new SARC service) | n = 15 | Qualitative interviews (Pre-operational stage) | 5 |
| | | Mixed stakeholders [SARC staff and other professional stakeholders (project lead, police, health, partner representatives] | n = 19 | Qualitative interviews (Post-operational stage) | |
| Robinson (2011)[7] [123] | SARCs, England and Wales, UK | SARC service users, staff and other stakeholders | n = 93 | Qualitative interviews | 4 |
| Ruch (1980)[8] [94] | Sexual assault treatment centre, Western US | SARC service users (adults using one Sexual Assault Treatment Centre), SARC staff and other professionals | n/s | Written questionnaires with all stakeholders and analysis of medical records | 1 |
| Schönbucher (2009)[9] [97] | Archway Glasgow SARC, Scotland, UK | Mixed stakeholders (SARC staff and Steering Group's members) | n = 33 | Online surveys or written questionnaires, and qualitative interviews | 5 |
| | | SARC service users | n = 23 | | |

*Note: Two studies were reported in two papers each (Cole 2007, 2008 and Campbell 2005, 2006). n/s = not specified.

[1] Fong (2016) compared experiences of caregivers of children using SARC like services (n = 12) and other sexual assault services (n = 10)

[2] Goddard (2015) reviewed the service pathways following sexual assault for children and young people in London, UK. The researchers interviewed professionals from various settings, including the Havens SARCs in London. The Havens accept referrals of adults and children and young people that have been victims of sexual assault.

[3] Harvey (2014) interviewed participants from various settings including the Amethyst and Ynys Saff SARCs; n for participants from each setting was not reported.

[4] Lippert (2018) explored experiences of caregivers experiences with children who attended therapy at the Child Advocacy Centre (n = 25) or who declined therapy (n = 20).

[5] Lovett (2004) included 49/56 service user interviews from SARC services and 110/136 professional stakeholder interviews from SARC service areas: the others were from comparison areas with no SARC provision. Only views from SARC stakeholder interviews are summarised here.

[6] Robinson (2009) conducted a process evaluation of the Ynys Saff SARC in Cardiff, Wales. Seven of the participants that were interviewed during the pre-operational phase were also interviewed in the post-operational phase.

[7] Robinson (2011) interviewed participants from 3 SARCs and 3 voluntary sector rape crisis settings; n for participants from each setting and participant group was not reported.

[8] Ruch (1980) report recommendations from structured surveys with several stakeholder groups: which groups' responses prompted each recommendation is not explicitly reported.

[9] Schonbucher (2009) conducted an evaluation of the pilot Archway Glasgow SARC, and interviewed staff in two phases between April 2007 and March 2009. Most staff were interviewed. twice. In the first phase, 23 SARC nurses and 10 SARC doctors were interviewed. In the second phase, 10 SARC nurses and 14 SARC doctors were interviewed. It is unclear how many Steering Group members were interviewed.

policy documents which summarised recommendations from the European Union-funded COSAI project [128–130], and one research project commissioned by the UK Home Office

**Table 4. Stakeholders' views—Recommendations for mental health and substance misuse provision in sexual assault services.**

| Area of service delivery | Recommendation | Studies (n) | Stakeholder group(s) |
|---|---|---|---|
| Screening/assessment for mental health or substance misuse needs | Equipment and medications to assess then treat mental health or substance misuse should be available in sexual assault services | 1 | Staff (Brooker 2018) [111] |
| | Some staff in sexual assault services want more training in assessing mental health and substance misuse problems to ensure appropriate referral on (especially with children (one study) and to identify PTSD (one study)) | 2 | Staff (Brooker 2018) [111] Mixed stakeholders (Musgrave 2014) [86] |
| | Staff want training to support people with intellectual disabilities in sexual assault services (including identifying ID, assessing capacity, best interest decision making and effective communication) | 1 | Staff (Olsen 2017) [90] |
| | Staff should assess service users' alcohol use at the point of presentation and at the time of the assault–to assess capacity to consent to sexual assault service procedures and assist forensic evidence-gathering | 2 | Mixed stakeholders (Cole 2008 [117], Musgrave 2014 [86]) |
| | Standardised mental health assessment tools and outcome measures should be used within Children's sexual assault services | 1 | Staff (Belew 2012) [37] |
| Provision of mental health or substance use support within sexual assault services | Immediate crisis counselling rated as the most important aspect of sexual assault service support to service users | 1 | Staff (Du Mont 2004) [54] |
| | Counselling should be provided within the sexual assault service | 4 | Staff (Belew 2012) [37] Mixed stakeholders (Lovett 2004 [20], Musgrave 2014 [86], Ruch 1980 [94]) |
| | A follow-up phone call from a counsellor 48–72 hours following visit to a sexual assault service was welcomed | 1 | Service users (Ericksen 2002) |
| | Trauma focused art therapy should be offered to children attending sexual assault services, especially those with difficulties vocally expressing their thoughts and feelings | 1 | Staff (Belew 2012) [37] |
| | Long-term, individually tailored therapy should be offered to children attending sexual assault services | 1 | Staff (Belew 2012) [37] |
| | Counselling offered in sexual assault services should not have an upper limit to the number of sessions | 1 | Mixed stakeholders (Lovett 2004) [20] |
| | Clients should be offered choice regarding the gender of their sexual assault service counsellor | 1 | Mixed stakeholders (Lovett 2004) [20] |
| | A 24/7 crisis line should be provided within a children's sexual assault service (e.g. for service users who are feeling suicidal) | 1 | Staff (Belew 2012) [37] |
| | Mentoring or peer support buddy schemes should be provided within sexual assault services to help clients' with wellbeing and practical support | 1 | Mixed stakeholders (Musgrave 2014) [86] |
| | Support workers should be service users' first point of contact with sexual assault services in to improve the take-up of follow-up support services | 1 | Staff (Schönbucher 2009) [97] |
| | Counselling support should be provided for as long as needed, given existing capacity | 1 | Staff and service users (Schönbucher 2009) [97] |
| | Sexual assault services should provide counselling/psychosocial support for children and young people until local services are available | 1 | Mixed stakeholders (Goddard 2015) [62] |
| | Sexual assault services should provide medical care and follow-up support for children under 13 years | 1 | Mixed stakeholders (Goddard 2015) [62] |
| Referral on from sexual assault services to mental health or substance misuse services | Sexual assault services should refer clients on to mental health services where indicated | 2 | Staff (Cowley 2014 [118]) Mixed stakeholders (COSAI 2012) [124] |
| | Service users would like to leave the sexual assault service with an appointment arranged for mental health follow-up (not just a referral made) | 1 | Service users (Ericksen 2002) [121] |
| | sexual assault services should make follow-up contact with clients to facilitate and encourage take-up of follow-on counselling | 3 | Staff (Maier 2012) [122] Mixed stakeholders (Lovett 2004) [20] Service users (Schönbucher 2009) [97] |
| | Flexibility is needed for follow-on mental health care to improve its accessibility to service users (e.g. re appointment times and location, funding travel to appointments) | 2 | Family caregivers (Fong 2016 [60], Lippert 2008 [78]) |
| | Integration is needed between sexual assault services and local counselling services, including mental health and drug and alcohol teams–including shared staffing and training (one study) and a dedicated single point of referral from sexual assault service to mental health teams and/or link workers (one study) | 2 | Staff (Campbell 1998) [113] Mixed stakeholders (Musgrave 2014) [86] |
| | Sexual assault services should have direct access to refer into MH services to avoid long delays for clients | 1 | Staff (Brooker 2015) [25] |
| | Partnership working with MH services is needed to improve their accessibility to service users | 1 | Staff (Brooker 2015) [25] |
| | Sexual assault services should provide service users with written, accessible information on follow-up services to encourage take-up | 1 | Service users (Schönbucher 2009) [97] |

**Table 5. Government and expert reports–characteristics.**

| Reference | Source | Description |
|---|---|---|
| COSAI (2012) [128] Policy briefing: comparing sexual assault interventions across Europe | European Union | EU funded and managed project which mapped sexual assault services in 34 countries and undertook 7 in-depth case studies to generate recommendations re sexual assault services |
| COSAI (2013) [129] UK guidance on sexual assault interventions | European Union | Brief report with UK-specific recommendations from the COSAI project |
| COSAI (2013b) [130] Comparing sexual assault interventions: summary report | European Union | Report summarising overall recommendations and findings from the seven case-studies (including England) from the COSAI project |
| Lovett (2004) [20] | Home Office, UK | Independent research study conducted for the Home Office, designed to provide recommendations and inform policy-making for SARCs. |
| ACPO (2004) [127] "Getting started guide" for SARCs | Association of Chief Police Officers (ACPO) Rape Working Group | Practical guidance for practitioners produced by the National SARCs steering group |
| DH (2005) [131] National service guidelines for developing SARCs | Department of Health, UK | National guidelines for commissioners of SARCs |
| DH (2009) [132] Revised national service guide: a resource for developing SARCs | Department of Health, UK | National recommendations for SARC service providers regarding minimum elements of SARC services |
| NHSE (2013) [133] Service Specification for SARCs No. 30 | NHS England | National service specifications for sexual assault services (typically SARCs) |
| NHSE (2015) [134] Commissioning framework for adult and paediatric SARC services | NHS England | National guidelines for commissioners of SARCs, including service standards, performance management indicators and care pathways |
| NHSE (2018) [22] Service Specification for SARCs No. 30 | NHS England | National guidelines for English SARCs, specifying service standards, performance management indicators and care pathways |

[20] specifically to help inform policy. These range in date from 2004 to 2018 and are listed in Table 5.

Recommendations from these ten policy documents regarding addressing mental health or substance use in SARCs are summarised in Table 6 below.

Some level of assessment of mental health problems for clients presenting at SARCs was advocated. Older UK guidance [132]recommended SARCs should undertake comprehensive assessment, including of mental health. Recent UK guidance [22, 134] says SARCs should assess risk of self-harm and vulnerability. However, the way that this should be done is not specified in any government guidance.

Regarding the provision of mental health support within SARCs, older UK guidance recommends that SARCs should provide in-house counselling (ideally not time-limited), SARC staffing should include counsellors and that SARCs should adopt a proactive approach to initiating and maintaining clients' take-up of counselling and support services [20, 131, 132]. By 2018 however, government guidance acknowledges that most SARCs *"are not designed to offer long-term support"*, so referrals on to primary care counselling services or voluntary sector services are required [22]. In England recent guidance [22, 134] suggests a threshold for severity of mental health needs–above IAPT [English primary care counselling services] level 3 – requiring a referral from a SARC to multidisciplinary input from community or acute mental health services. What role SARCs should have in providing support for clients with mental health needs at or below this severity threshold is not specified. There is an absence of guidance in government policy documents about addressing clients' needs regarding drug or alcohol use in SARCs.

Recommendations from the COSAI 2012 project [128–130] were congruent with English national policy guidance, but were notably more specific in directing SARCs to focus actively on engaging people with mental health problems, and on ensuring provision of CBT therapies to treat clients with symptoms of PTSD. COSAI guidance also advocated provision of

**Table 6. Government and expert reports—recommendations for mental health and substance misuse provision in UK SARCs.**

| Area of SARC service delivery | Recommendation | Sources |
|---|---|---|
| Screening/assessment for mental health or substance misuse needs | Medical assessment should include assessment for "self-harm and vulnerability" | DH 2009 [132]*; NHSE 2015 [134]; NHSE 2018 [22] |
| | A comprehensive assessment includes mental health | DH 2009 |
| | SARCs should focus special attention on engaging people with learning difficulties or mental health problems | COSAI 2013 [129] |
| Provision of mental health or substance use support within the SARC | Crisis workers will provide emotional and practical support throughout a service user's time in a SARC | NHSE 2015 [134] |
| | SARC counselling and support should cover a range of needs, including: initial crisis care, informal support and advocacy, and longer-term therapeutic support. | Lovett 2004 [20] |
| | Design and provision of sexual assault services should build in providing cognitive behavioural therapies to treat symptoms of post-traumatic stress disorder | COSAI 2012 [128] |
| | SARC staff should be proactive in initiating and maintaining client contact with counselling and support services | Lovett 2004 [20] |
| | SARCs should offer counselling. Provision of mental health services within the SARC increases "likelihood the client will access the treatment they need" | COSAI 2012 [128]; DH 2009 [132] |
| | SARC staffing should include counsellors/psychologists | DH 2009 [132] |
| | Example care pathway diagram includes in-house SARC counselling offered pre-trial of "6–10 sessions or as appropriate" | NHSE 2013 [133]; NHSE 2015 [134]; NHSE 2018 [22] |
| | Means to provide support/counselling to family or close friends of the SARC client is desirable, where required | COSAI 2013b [130] |
| Referral on from SARC to mental health or substance misuse services | SARCs will ensure immediate access to mental health crisis services as needed | DH 2009 [132]; NHSE 2015 [134]; NHSE 2018 [22] |
| | SARCs should be integrated within wider care systems and have established care pathways to NHS and voluntary sector mental health services (e.g. IAPT), including specialist children's services | DH 2009 [132]; NHSE 2013 [133]; NHSE 2015 [134]; NHSE 2018 [22] |
| | SARCs should inform service users about psychological therapies and independent advocacy | NHSE 2015 [134]; NHSE 2018 [22] |
| | SARCs should provide referrals to psychological therapies and counselling services | NHSE 2013 [133]; NHSE 2015 [134]; NHSE 2018 [22] |
| | For needs greater than IAPT level 3 support, SARCs should refer to local community mental health services | NHSE 2015 [134]; NHSE 2018 [22] |
| | Access to specialist counselling and support for LGBT SARC clients is desirable | COSAI 2013b [130] |
| | SARCs should monitor access to and take-up of pre-trial counselling and therapy | COSAI 2012 [128]; NHSE 2015 [134]; NHSE 2018 [22] |
| | Follow-up rates/severity of PTSD for SARC clients is an appropriate SARC quality indicator | COSAI 2012 [128] |

* DH 2009 guidance states that it incorporates and updates guidance from 2005 DH national service guidelines and "Getting Started Guide" for SARCs, which are therefore not summarised separately in the table.

counselling support for families and close friends of the SARC client where needed, and access to specialist counselling support for LGBT SARC clients.

## Discussion

### Summary of findings

Our synthesis of 107 included papers indicates a lack of consensus on how to identify mental health and substance use needs in sexual assault services, with identification based more often on unanchored clinical judgement than informed by standardised tools. Some sexual assault services offer in-house mental health support, and this tends to be supportive counselling rather than specific structured psychological interventions delivered by a qualified and accredited therapist. Where papers described referring to other services most did not offer an active follow-up to promote service users' engagement with other helping agencies. In general, child advocacy centres (US, Turkey) and sexual assault services in Nordic countries were most likely to provide structured mental health support (such as using validated assessment measures, in-house structured psychological interventions, and referral with active follow-up).

However, it is important to recognise that many of the papers in the review were not specifically written to focus on mental health, and so provided very limited detail about what sexual assault services do to address mental health and substance misuse. This may reflect that mental health and substance misuse are not seen as core areas of activity for sexual assault services, compared to their defined roles in forensic evidence gathering and physical healthcare.

Our review found only five studies evaluating psychological interventions for people attending sexual assault services that provided mixed evidence of moderate quality. Four of these reported on brief psychoeducational VIs and one on a brief six session "cognitive processing therapy". It should be noted that we did not exclude papers of low quality and our review therefore indicates there is currently no robust empirical evidence to guide sexual assault services in how to address service users' mental health and substance misuse difficulties.

Stakeholders valued accessibility, flexibility, continuity of care, in-house psychological support, and staff trained in mental health and specialist support for LGBT people and people with learning difficulties. Service users specifically highlighted the need for longer-term support and/or follow-up including referral, integration and partnership between sexual assault services and local counselling services. Although views from different types of stakeholder groups did not conflict, there was relatively little evidence on service users' views. Despite the existence of evidence based therapies for mental health problems following sexual assault, there is a lack of specificity in UK government and policy guidance about how SARCs should be best implemented to address service users' mental health and substance needs.

Despite the high prevalence of substance misuse issues in people who have been sexually assaulted [141] we found that this issue was barely mentioned. In sum, despite the very high prevalence of mental health and substance misuse issues amongst those who experience sexual assault, current evidence is very limited in how far it can guide UK and international sexual assault services in how best to assess and address these needs. There is wide variation in approaches both across and within sexual assault service models.

### Strengths and limitations

To our knowledge, this is the first systematic review to collate all types of relevant evidence regarding approaches to addressing mental health and substance misuse needs in sexual assault services internationally. Due to limited resources, the web-based search for policy guidance was limited to England and papers without a version in English were excluded. As such,

we may have missed some relevant papers that could have improved our understanding of international variations in current and optimal models of mental health care in sexual assault services. However, our otherwise comprehensive search and selection strategy minimises publication and reporting bias [142].

Due to the limited information provided by most papers, we were only able to describe approaches of mental health provision in sexual assault services in very broad categories that may mask large variations in care. Findings should therefore be considered with caution. Since we included studies from 1979 onwards, our findings may not reflect current practice and stakeholder perspectives in all areas. Furthermore, some qualitative studies with mixed samples were excluded if the specific views of stakeholders from sexual assault stakeholders could not be distinguished from others. As a result, some potentially relevant evidence may have been excluded.

We used the MMAT quality assessment tool because it can assess studies across a wide range of different designs and methodologies. However, the tool is relatively crude. As a result, studies were able to obtain high quality ratings despite significant shortcomings (e.g. lack of clearly specified primary outcomes or published protocol). Due to limited resources, a single researcher conducted quality assessments.

Finally, implications for practice from our review are limited by the quality and quantity of available evidence from included studies. The few empirical studies we found had important methodological weaknesses including potentially inadequate sample sizes and high attrition rates. Relatively few qualitative studies explored service users' views. We found almost no information about how substance misuse needs should be assessed and addressed in sexual assault services.

## Implications for research

We have identified three domains of mental health care (identification, provision of in-house support, and referral onwards to external services), where different approaches can be distinguished. This can assist future researchers in describing and evaluating sexual assault services. However, further clarification and distinction between different counselling approaches in sexual assault services is required as these are poorly described in the literature to date.

There is an urgent need for high quality evidence to inform service development. Understanding which models of service delivery and organisation are most effective in supporting access to effective mental health and substance misuse treatment where required is an area of high importance. Specific key questions include: i) does structured screening of mental health and substance misuse in sexual assault services increase the uptake of mental health and substance use support, and improve outcomes? ii) Which psychological therapies should be provided by sexual assault services or by external services and what are the referral criteria? These topics could be explored through randomised controlled trials (RCTs) or naturalistic comparisons.

There is, also, a need for research that seeks to explore service users' views of how they would like their mental health and/or substance use needs to be addressed in sexual assault services and the acceptability of current service provision. It is unclear for instance, whether service users want to return to the sexual assault service for counselling support, given that the medical forensic examination was undertaken there. Future research should also seek the views of seldom-heard groups including people from black and ethnic minorities, LGBT groups, and people with learning difficulties.

## Implications for policy and practice

The variation in mental health and substance misuse service provision in sexual assault services creates an element of "postcode lottery" for service users. Clearer guidance for service

planners and commissioners, informed by robust evidence about optimal service organisations and pathways, is required.

If sexual assault services are to be able to help people access support for mental health or substance use (whether provided within the service or via referral on), they first need to be able to identify people with mental health and substance misuse needs. However, our review suggests that most sexual assault services do not have consistent, robust means to do this. There is a clear need for routine, structured screening, using reliable measures, as advocated by experts in the field [25, 141]. The timing of assessments may also be important, as not all mental health problems can be reliably assessed immediately following a traumatic event. For instance, the PTSD diagnosis is given if one's symptoms last longer than a month [15]. As a result, follow-up screening may also be needed.

Our review also suggests that most sexual assault services are not well equipped to help service users access effective treatments for mental health or substance misuse, even where difficulties have been identified. For example, the most common mental health problem following sexual assault is PTSD [7]. Recommended treatments for PTSD are structured psychological therapies (trauma-focused CBT or EMDR) to be provided promptly if symptoms persist for more than one month [15]. However, almost no papers in our review reported SARCs providing these treatments, and only a small minority of sexual assault services were reported to provide any assertive follow-up to ensure service users who need it can access treatment from other specialist services. In circumstances like these, SARCs may not act as a "one-stop shop" to coordinate care as intended. Stakeholders advocated for good communication between sexual assault services and other agencies, including mental health teams: as well as facilitating referral pathways, this may provide an opportunity for SARCs to raise awareness of the prevalence and effects of sexual assault among mental health staff.

Overall, despite all the areas of uncertainty that remains following this review, there is an identified consensus from stakeholders that SARCs should be universal and include: an accessible 24/7 service, which can meet all of people's immediate needs, treat people with courtesy and kindness, avoiding repeat assessments and the need to retell the story of the assault as far as possible, providing information about options and establishing that nothing will be done in the SARC without the service user's explicit consent, and providing female staff to female service users. These ways of working constitute an essential minimum for SARCs, whatever their service model, to respond with care and kindness, promote recovery and avoid additional distress for people seeking help following a sexual assault.

## Supporting information

**S1 File.**
(DOC)

**S2 File.**
(DOCX)

**S3 File.**
(DOCX)

**S4 File.**
(DOCX)

**S5 File.**
(DOCX)

**S6 File.**
(PDF)

## Acknowledgments

This paper was written as part of the MiMoS study. We would like to thank the study steering committee for their contribution and continued support.

## Author Contributions

**Conceptualization:** Elizabeth Hughes, Brynmor Lloyd-Evans.

**Data curation:** Theodora Stefanidou, Elizabeth Hughes, Katherine Kester.

**Formal analysis:** Theodora Stefanidou, Katherine Kester, Brynmor Lloyd-Evans.

**Funding acquisition:** Elizabeth Hughes, Steven Ariss, Charlie Brooker, Gail Gilchrist, Mike Lucock, Kylee Trevillion, Brynmor Lloyd-Evans.

**Investigation:** Theodora Stefanidou, Elizabeth Hughes, Katherine Kester, Amanda Edmondson, Rabiya Majeed-Ariss, Christine Smith, Steven Ariss, Charlie Brooker, Gail Gilchrist, Sarah Kendal, Mike Lucock, Fay Maxted, Concetta Perot, Rebekah Shallcross, Kylee Trevillion, Brynmor Lloyd-Evans.

**Methodology:** Theodora Stefanidou, Elizabeth Hughes, Brynmor Lloyd-Evans.

**Project administration:** Theodora Stefanidou, Elizabeth Hughes, Brynmor Lloyd-Evans.

**Supervision:** Brynmor Lloyd-Evans.

**Writing – original draft:** Theodora Stefanidou, Elizabeth Hughes, Brynmor Lloyd-Evans.

**Writing – review & editing:** Theodora Stefanidou, Elizabeth Hughes, Katherine Kester, Amanda Edmondson, Rabiya Majeed-Ariss, Christine Smith, Steven Ariss, Charlie Brooker, Gail Gilchrist, Sarah Kendal, Mike Lucock, Fay Maxted, Concetta Perot, Rebekah Shallcross, Kylee Trevillion, Brynmor Lloyd-Evans.

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
