## [Decision Letter · Decision Letter 0]

22 Jan 2020

PONE-D-19-28688

The identification and treatment of mental health and substance misuse problems in Sexual Assault Referral Centres (SARCs): A systematic review

PLOS ONE

Dear Dr Lloyd-Evans,

Thank you for submitting your manuscript to PLOS ONE. After careful consideration, we feel that it has merit but does not fully meet PLOS ONE’s publication criteria as it currently stands. Therefore, we invite you to submit a revised version of the manuscript that addresses the points raised during the review process.

We would appreciate receiving your revised manuscript by 15/03/2020. To enhance the reproducibility of your results, we recommend that if applicable you deposit your laboratory protocols in protocols.io, where a protocol can be assigned its own identifier (DOI) such that it can be cited independently in the future. For instructions see: http://journals.plos.org/plosone/s/submission-guidelines#loc-laboratory-protocols

We look forward to receiving your revised manuscript.

Kind regards,

Karen-Leigh Edward

Academic Editor

PLOS ONE

Reviewers' comments:

Reviewer's Responses to Questions

**Comments to the Author**

1. Is the manuscript technically sound, and do the data support the conclusions?

Reviewer #1: Partly

Reviewer #2: Yes

2. Has the statistical analysis been performed appropriately and rigorously? 

Reviewer #1: N/A

Reviewer #2: N/A

3. Have the authors made all data underlying the findings in their manuscript fully available?

Reviewer #1: Yes

Reviewer #2: Yes

4. Is the manuscript presented in an intelligible fashion and written in standard English?

Reviewer #1: Yes

Reviewer #2: Yes

5. Review Comments to the Author

Reviewer #1: Major comments:

Abstract: The fact that stakeholders and guidelines also recommended referral to and partnership with local counselling services should be mentioned since these recommendations are essential.

Introduction

The reader may get the impression that suggestions from stakeholders and national guidelines give correct and complete recommendations. A description of evidence based methods is lacking, although mentioned in the discussion. After the section on sexual violence in the introduction, present evidence based recommendations on how to identify, follow up and treat mental illness and substance use following sexual violence should be stated/exemplified: standardised screening tools to identify acute stress disorder, PTSD, depression, anxiety, substance abuse disorders, and risk of suicide and examples of structured psychological therapies.

Aims

The second aim should clarify that the study included routines for referral and follow up.

Discussion

The comments (line 524) on the stakeholder should include their mentions of the importance of referral, integration and partnership between SARCs and local counselling services for mental health and substance misuse.

The conclusion (line 529) that ”the lack of specificity in government and policy guidance …. may reflect the lack of available evidence” is doubtful since evidence based therapies exist.

Implications for research

i) SARCs can increase the uptake of mental health ……..

ii) The suggestion to study provision of mental health support provided by SARCs contra routine referral to external services depends of what kind of support that is provided. The question is rather, which therapies should be done within SARCs and by external services respectively and to define criteria for referral.

The last section, line 608-16 seems less relevant. A reflection on the role of psychiatric and substance abuse treatment clinics on the treatment of victims of sexual violence may be of interest. Is there a sufficient awareness among psychiatrists of the long term effects of sexual violence? Are patients with mental illness and/or substance misuse routinely asked about experience of sexual violence? There is a need of common knowledge development.

Minor comments:

Title: In the title change SARCs to Sexual Assault Services since the designation SARC is specific for UK (also commented under AIMs)

Aims

In order to know when the paper discusses SARCs in UK or includes corresponding units internationally it would help to name them SARCs and “sexual assault services" respectively.

Introduction

Long-term effects of sexual violence: Sex as self‑injury should be included since it is a specific form of self-harm and the risk of re- victimization needs to be mentioned. Sexual violence is also associated with physical illness.

Reviewer #2: This is a well-written systematic review addressing the identification and treatment of mental health and substance misuse problems in Sexual Assault Referral Centres (SARC). The authors have adhered to recommended methodology for SRs and described their methods clearly. The conclusions appear robust and the recommendation in the discussion on future research is clear and helpful.

There are a few points to make:

1. The abstract does not make clear that there was very little evidence found regarding interventions or support for substance misuse. They refer to 'post-rape psychopathology' but the absence of consideration of drug misuse (one of the key objectives in the review) should be highlighted in the abstract.

2. For question 2, they have not excluded any papers on quality grounds. While this may be less of an issue if there is no attempt at metanalysis, the inclusion of low quality papers needs to be qualified by appropriate levels of emphasis, lest erroneous conclusions be drawn. This could be made clear.

3. There is comment made in the discussion about whether SARCs are the right setting to provide follow-up for either substance misuse or other mental health issues. I could not see if this issue addressed in the results, unless I missed it. It would be important to highlight this.

There are few typos etc:

SANE is not defined in the text when is appears (except via the Box).

Line 572 - delete 'can'

Table 1 - footnotes to this table should include a explanation of the acronyms

---

## [Author Response · Author response to Decision Letter 0]

13 Mar 2020

Dear reviewers, 

Thank you very much for your time and feedback on our manuscript. The points you have raised were very helpful in revising and improving our paper. We have been able to incorporate changes to reflect most of your suggestions. Please see below a point-by-point response to your comments and suggestions:

Comments from reviewer #1

1. Abstract: The fact that stakeholders and guidelines also recommended referral to and partnership with local counselling services should be mentioned since these recommendations are essential. 

We agree. We have revised the abstract accordingly (lines 67-69)

2. Introduction: The reader may get the impression that suggestions from stakeholders and national guidelines give correct and complete recommendations. A description of evidence based methods is lacking, although mentioned in the discussion. After the section on sexual violence in the introduction, present evidence based recommendations on how to identify, follow up and treat mental illness and substance use following sexual violence should be stated/exemplified: standardised screening tools to identify acute stress disorder, PTSD, depression, anxiety, substance abuse disorders, and risk of suicide and examples of structured psychological therapies. 

Thank you for pointing this out. We agree with this comment and have added a paragraph summarising evidence based recommendations (lines 112-120). 

3. The second aim should clarify that the study included routines for referral and follow up. 

We have amended the second aim to clarify that we sought available evidence for the effectiveness of all aspects of service delivery and organisation in SARCs. 

4. Discussion: The comments (initial manuscript line 524) on the stakeholder should include their mentions of the importance of referral, integration and partnership between SARCs and local counselling services for mental health and substance misuse. 

We had tried to capture these in continuity of care and the need for longer-term support and follow-up. Changes have been applied to make these points clearer (lines 574-575)

5. Discussion: The conclusion (initial manuscript line 529) that ”the lack of specificity in government and policy guidance …. may reflect the lack of available evidence” is doubtful since evidence based therapies exist. 

Thank you for raising this. We were referring to the lack of empirical evidence about how SARCs should be best implemented to address people’s mental health and substance use needs (In-house psychological support; referral to other services etc.). We agree with the reviewer that evidence is available about effective treatments for people experiencing mental health problems following sexual assault and have revised the sentence accordingly (lines 577-580).

6. Implications for research: 

i) SARCs can increase the uptake of mental health… 

We have added a sentence to the research implications (lines 624-626) to acknowledge that understanding how SARCs can best promote access to and uptake of effective mental health care where required is of key importance.

ii) The suggestion to study provision of mental health support provided by SARCs contra routine referral to external services depends of what kind of support that is provided. The question is rather, which therapies should be done within SARCs and by external services respectively and to define criteria for referral. 

We agree and have incorporated the suggestion in the manuscript (lines 630-631).

iii) The last section, initial manuscript line 608-16 seems less relevant. A reflection on the role of psychiatric and substance abuse treatment clinics on the treatment of victims of sexual violence may be of interest. Is there a sufficient awareness among psychiatrists of the long term effects of sexual violence? Are patients with mental illness and/or substance misuse routinely asked about experience of sexual violence? There is a need of common knowledge development. 

It was notable in the qualitative findings that stakeholders talked a lot about generic accessibility. We think it is important to stress these points and we would like to keep this section. We have added a sentence to reinforce that stakeholders advocated good communication and links between SARCs and other agencies, including mental health services, and that this provides opportunities to raise awareness about the prevalence and effects of sexual violence among mental health staff (lines 665-668). 

 Minor comments:

7. Title: In the title change SARCs to Sexual Assault Services since the designation SARC is specific for UK (also commented under AIMs). 

We agree. Change applied. 

8. Aims: In order to know when the paper discusses SARCs in UK or includes corresponding units internationally it would help to name them SARCs and “sexual assault services" respectively. 

We agree that using SARC as a generic term AND to describe a specific service model might be confusing. We have replaced SARCs with “sexual assault services” where necessary throughout the text, tables and appendices. We’ve also edited the abstract so it matches the new use of terms (lines 45-47).

9. Introduction: Long-term effects of sexual violence: Sex as self‑injury should be included since it is a specific form of self-harm and the risk of re- victimization needs to be mentioned. Sexual violence is also associated with physical illness. 

We agree. We have incorporated your suggestion in the introduction (lines 103, 109-111).

Comments from reviewer #2

1. The abstract does not make clear that there was very little evidence found regarding interventions or support for substance misuse. They refer to 'post-rape psychopathology' but the absence of consideration of drug misuse (one of the key objectives in the review) should be highlighted in the abstract. 

We agree. Manuscript revised (lines 61-2).

2. For question 2, they have not excluded any papers on quality grounds. While this may be less of an issue if there is no attempt at metanalysis, the inclusion of low quality papers needs to be qualified by appropriate levels of emphasis, lest erroneous conclusions be drawn. This could be made clear. 

Thank you for raising this. We agree and have emphasised this point in the conclusion (line 568).

3. There is comment made in the discussion about whether SARCs are the right setting to provide follow-up for either substance misuse or other mental health issues. I could not see if this issue addressed in the results, unless I missed it. It would be important to highlight this. 

We agree our speculation about whether service users would prefer to receive follow-up support in SARCs or elsewhere was not directly rooted in the results, so we have deleted this comment. 

4. There are few typos etc: all addressed

1. SANE is not defined in the text when is appears (except via the Box). Defined (lines 146-7).

2. Line 572 - delete 'can'. Deleted.

3. Table 1 - footnotes to this table should include a explanation of the acronyms. Agree. Footnotes explaining the acronyms have been added.

We look forward to hearing from you in due time regarding our submission and to respond to any further questions and comments you may have.

Yours Sincerely,

Bryn Lloyd-Evans

---

## [Editor Report · Decision Letter 1]

20 Mar 2020

The identification and treatment of mental health and substance misuse problems in Sexual Assault Services: A systematic review

PONE-D-19-28688R1

Dear Dr.Lloyd-Evans,

We are pleased to inform you that your manuscript has been judged scientifically suitable for publication and will be formally accepted for publication once it complies with all outstanding technical requirements.

With kind regards,

Karen-Leigh Edward

Academic Editor

PLOS ONE

---

## [Editor Report · Acceptance letter]

25 Mar 2020

PONE-D-19-28688R1 

The identification and treatment of mental health and substance misuse problems in Sexual Assault Services: A systematic review 

Dear Dr. Lloyd-Evans:

I am pleased to inform you that your manuscript has been deemed suitable for publication in PLOS ONE. Congratulations! Your manuscript is now with our production department. 

With kind regards,

on behalf of

Professor Karen-Leigh Edward 

Academic Editor

PLOS ONE